# Reconstructing Heterogeneous Biomolecules via Hierarchical Gaussian Mixtures and Part Discovery

**Shayan Shekarforoush**[1,2] **David B Lindell**[1,2] **Marcus A Brubaker**[1,2,3] **David J Fleet**[1,2]
[1]University of Toronto    [2]Vector Institute    [3]York University
{shayan,lindell,fleet}@cs.toronto.edu   mab@eecs.yorku.ca

## Abstract

Cryo-EM is a transformational paradigm in molecular biology where computational methods are used to infer 3D molecular structure at atomic resolution from extremely noisy 2D electron microscope images. At the forefront of research is how to model the structure when the imaged particles exhibit non-rigid conformational flexibility and compositional variation where parts are sometimes missing. We introduce a novel 3D reconstruction framework with a hierarchical Gaussian mixture model, inspired in part by Gaussian Splatting for 4D scene reconstruction. In particular, the structure of the model is grounded in an initial process that infers a part-based segmentation of the particle, providing essential inductive bias in order to handle both conformational and compositional variability. The framework, called CryoSPIRE, is shown to reveal biologically meaningful structures on complex experimental datasets, and establishes a new state-of-the-art on CryoBench, a benchmark for cryo-EM heterogeneity methods. Project Webpage.

## 1 Introduction

Single-particle cryo-electron microscopy (cryo-EM) is a computationally driven experimental paradigm that is transforming molecular biology by enabling 3D structure determination of biomolecules, such as proteins and viruses, at near-atomic resolutions [3, 18, 38]. The core computational task is estimating a 3D structure from 2D images with unknown orientation and position, under extremely low signal-to-noise conditions. Essential to their biological function, biomolecules exhibit varying degrees of *conformational flexibility*, where structures deform non-rigidly, and *compositional variation*, where parts of a structure may be present in some images and absent in others (see Fig. 1). Accordingly, a major challenge in cryo-EM is the estimation of 3D structures from such heterogeneous data and, to that end, how to infer meaningful representations of structures such as parts that capture their heterogeneity. The crux of this challenge is how to effectively represent and regularize this variability without overfitting to the noise in cryo-EM images. Existing methods, while encouraging, are generally limited in either expressiveness, interpretability, or efficiency.

Here, we propose CryoSPIRE, a new method for heterogeneous reconstruction. We leverage a part-based Gaussian mixture model (GMM) of 3D density that enables CryoSPIRE to represent both conformational and compositional heterogeneity, unlike some existing deformation-based methods [13, 33]. Further, it provides a naturally interpretable and physically plausible, part-based structure in contrast to existing latent variable methods based on linear density subspaces [10, 32] or neural field models [19, 20, 47]. A key challenge with part-based GMMs concerns initialization and the discovery of parts. We propose a novel method for part discovery which estimates a coarse-grained GMM with per-Gaussian learnable features (c.f., [2]) and an MLP which defines Gaussian locations and amplitudes. We show that these learned features naturally encode characteristics of structural heterogeneity, which we leverage to infer a part-based segmentation of the structure. Inspired in part by Scaffold-GS [21], we define CryoSPIRE (Scaffold Part-Aware Mixture of Gaussians), a hierarchical model which estimates a Gaussian mixture wherein the composition of components

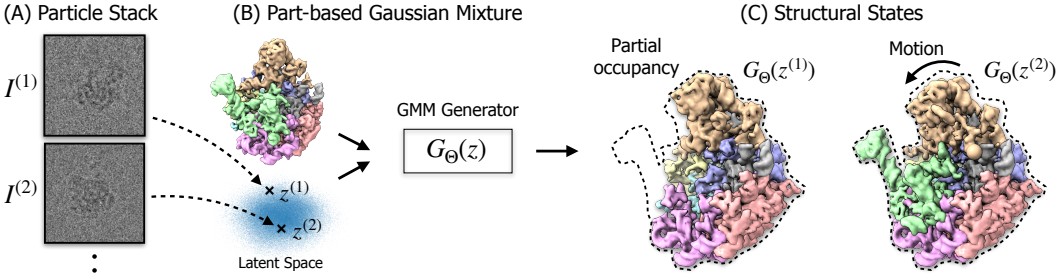

Figure 1: (A) Based on a stack of noisy particle images, (B) CryoSPIRE learns a part-based Gaussian mixture, with parameters $\Theta$, and a latent space representing structural heterogeneity. Given a latent code $z$, a generator produces a 3D density map. (C) The model supports compositional variability (e.g., $G_\Theta(z^{(1)})$ with a missing part), and conformational flexibility (e.g., $G_\Theta(z^{(2)})$ with part deformation).

and their deformation are defined in terms of a set of anchors, corresponding to parts. The resulting model naturally allows for the arbitrary combination of parts which can both rigidly move and locally deform as a function of an input heterogeneity latent code (see Fig. 1).

To our knowledge, this is the first GMM-based model to be successfully benchmarked on CryoBench [15], a standardized benchmark for cryo-EM heterogeneity with ground-truth labels. In particular, CryoSPIRE outperforms widely used and state-of-the-art methods [10, 19, 32, 33, 47], sometimes by a wide margin. Through ablations, we also validate key design choices, demonstrating the benefits of Gaussian features over positional encoding as in DynaMight [40], and highlighting the benefits of hierarchical motion modeling. Finally, on experimental data, CryoSPIRE automatically discovers representations of 3D density maps that correspond to biologically meaningful parts.

To summarize our contributions: we propose a new method enabling part-discovery on 3D biomolecular structures based on a coarse-grained GMM. This part-based structure is used to initialize a novel, hierarchical GMM-based model for heterogeneous reconstruction with compositional and conformational variability. The resulting framework, CryoSPIRE, establishes a new state-of-the-art on quantitative benchmarks and qualitative experimental datasets.

## 2  Background and Related Work

**Latent Variable Models.** Heterogeneous cryo-EM reconstruction methods typically introduce latent variables to represent structural variability of the 3D density map. 3DVA [32] and RECOVAR [10] learn a linear subspace to represent variation in 3D density maps, with clever numerical and regularization techniques to optimize high-dimensional basis maps at high spatial resolutions. Nevertheless, to model large-scale continuous motion with a high dimensional subspace, memory requirements are prohibitive. Much current work has shifted to nonlinear latent models and deep learning [14, 19, 47], with Cryo-DRGN [47] and DRGN-AI [19] using auto-encoders to obtain latent codes and conditional coordinate networks [24] to generate density maps. Such latent-variable models are hard to interpret, however, as conformational and compositional heterogeneity are not decoupled, and they provide no explicit model of motion between conformational states. By contrast, the latents in 3DFlex [33] encode flow fields that model the conformational deformation of a canonical structure. While resolving detailed motion and improving the quality of density maps, 3DFlex cannot handle compositional heterogeneity, and it is highly sensitive to regularization, often requiring substantial trial and error.

**GMM-Based Methods.** Gaussian mixtures have been used to model 3D density [4, 5, 6, 40]; they provide a sparse, compact representation in which conformation and compositional variability are modeled in terms of positions and amplitudes of Gaussian components. With Gaussian components viewed as atomic primitives, such models also facilitate physics-based priors [6, 40] and subsequent molecular model fitting. Nevertheless, existing GMM-based methods fall short in various ways. E2GMM [4] and related methods [5, 6] generate GMM parameters with a single network, which scales poorly to large numbers of Gaussians. Further, their multi-scale smoothness priors [6] are based on an arbitrary hierarchy which fails to capture part-based structures, thus resorting to manual part masks to resolve and estimate local motions. DynaMight [40] is similar to CryoSPIRE in defining

an explicit motion model, but it is unable to handle compositional variations, and, as we show, its positional encodings are inferior to our learnable features.

**Gaussian Splatting.** Beyond cryo-EM, the effectiveness of GMMs has been demonstrated in 3D Gaussian Splatting [16, 48], a technique which provides a fast approximation to the volume rendering integral [8, 23], enabling efficient high-fidelity reconstruction of 3D scenes from multi-view images [11, 17, 22, 44, 45, 46]. 3D Gaussian Splatting represents scene appearance and structure using thousands to millions of Gaussian components, each associated with parameters that control opacity and view-dependent color. CryoSPIRE is in part inspired by Gaussian Splatting [2, 21], but tailored to cryo-EM, with a different image formation model, images with signal-to-noise ratios less than 5%, and a novel method for part discovery.

**GMM Image Formation.** Following [4, 5, 6, 40], we parameterize the terms of a Gaussian mixture with center $\boldsymbol{c} \in \mathbb{R}^3$, isotropic scale $s \in \mathbb{R}$, and an amplitude $m \in \mathbb{R}$:

$$f(\boldsymbol{p}) \;=\; \sum_i m_i \exp\left(-\frac{||\boldsymbol{p} - \boldsymbol{c}_i||_2^2}{2s_i^2}\right) \;, \tag{1}$$

for location $\boldsymbol{p} \in \mathbb{R}^3$. We transform the GMM into the observation space for the $n$-th particle image, with a rotation $\boldsymbol{R}^{(n)} \in SO(3)$ and translation $\boldsymbol{t}^{(n)} \in \mathbb{R}^3$, followed by an integral projection along the $z$-axis of the microscope, to obtain a noise-free 2D image, $\tilde{I}(\tilde{\boldsymbol{p}})$, [4]:

$$\tilde{I}^{(n)}(\tilde{\boldsymbol{p}}) \;=\; \sum_i \sqrt{2\pi} s_i m_i \exp\left(-\frac{|| \tilde{\boldsymbol{p}} - [\boldsymbol{R}^{(n)}\boldsymbol{c}_i + \boldsymbol{t}^{(n)}]_{xy} ||_2^2}{2s_i^2}\right) \;, \tag{2}$$

where $\tilde{\boldsymbol{p}} \in \mathbb{R}^2$ and $[\cdot]_{xy}$ is an operator to discard $z$ coordinate of the input position. Cryo-EM images are then convolved with microscope point spread function and corrupted by additive mean-zero Gaussian noise, $\hat{I}^{(n)} = g^{(n)} \star \tilde{I}^{(n)} + \epsilon^{(n)}$. Like other cryo-EM models, the parameters are typically optimized by minimizing a squared L2 reconstruction loss between model predictions and observed images. See the supplement for more details on image formation and the image likelihood.

# 3  CryoSPIRE

Heterogeneous cryo-EM involves non-rigid 3D reconstruction from noisy 2D images. For such an inverse problem, regularization and inductive bias are key. Local smoothness is a natural choice for regularization, however, smoothness alone is not sufficient as nearby regions can deform in somewhat independent ways [33]. Further, the presence or absence of biomolecule parts is not dictated by spatial proximity alone. Macromolecular complexes, like many objects, naturally possess a part-based structure that connects to their compositional and conformational variations. But a coherent 3D part-decomposition is unavailable *a priori*, and estimating parts from noisy 2D observations is inherently challenging. As a consequence, prior work resort to manually designed masks or meshes [25, 33].

Here, we propose a novel two-stage GMM-based framework. Given particle images with corresponding poses $\{(I^{(n)}, \boldsymbol{R}^{(n)}, \boldsymbol{t}^{(n)})\}_{n=1}^N$, and a crude initial 3D structure, we first optimize a coarse-grained GMM in which each Gaussian component is augmented with a learnable feature vector (c.f., [2]). We observe that the learned features encode meaningful information about structural regularities. In particular, Gaussian components that coherently deform or consistently appear or disappear receive similar features, facilitating the inference of a part-based segmentation of the particle. Second, based on the identified parts and inspired by Scaffold-GS [21], we define a part-aware Gaussian mixture model in terms of a set of anchors, one per part, each with a corresponding set of Gaussians. Optimizing this representation recovers a high-resolution representation of 3D density maps with compositional and conformational variability. In what follows, we describe the part-based hierarchical model, (Fig. 2B–D), followed by the part discovery method and initialization scheme (Fig. 2A).

## 3.1  Part-Aware Gaussian Mixture

We first specify the form of the part-aware latent-conditioned mixture model; Table 1 provides a summary of the notation used. The model is conditioned on a latent coordinate $\boldsymbol{z} \in \mathfrak{Z} \subset \mathbb{R}^D$ for each image, which specifies the state of the macromolecule. The density model itself comprises a set of anchors, each associated with a meaningful part of the macromolecule (Fig. 2B). We parameterize the

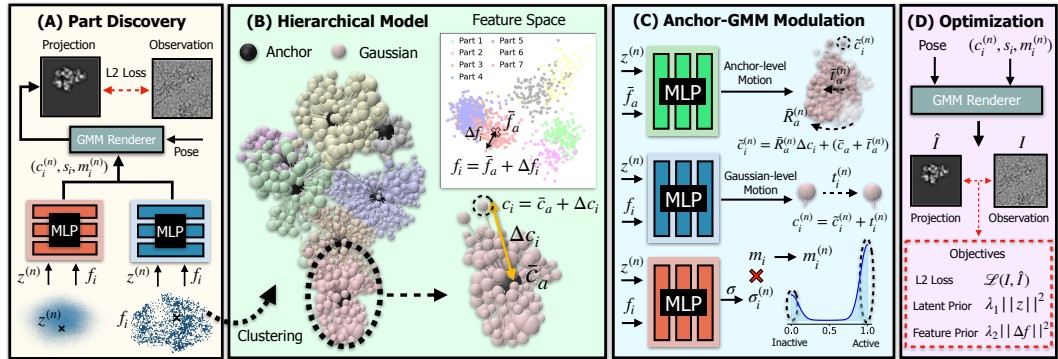

Figure 2: Overview of CryoSPIRE. **(A)** To infer parts, we optimize a coarse GMM with neural networks that generate Gaussian amplitudes and centers, conditioned on image latent codes and Gaussian features. **(B)** Clustering on learned Gaussian features provides meaningful parts. The CryoSPIRE model comprises one anchor and a set of Gaussians per part. **(C)** MLPs generate the rigid-body motion of each anchor (top), per-Gaussian displacements relative to the anchor frames (middle), and per-Gaussian activations in (0,1) to represent occupancy (bottom). **(D)** A reconstruction loss compares observed images to 2D projection of the corresponding 3D GMM. Priors encourage bounded latent code and small feature offsets.

| Gaussians | | Anchors | | Particles | |
|---|---|---|---|---|---|
| $i$ | Gaussian index | $a$ | Anchor index | $n$ | Particle index |
| $\mathbf{c}_i$ | Gaussian center | $a_i$ | Anchor index of $i$-th Gaussian | $I^{(n)}$ | Observed image |
| $\mathbf{m}_i$ | Gaussian amplitude | $\bar{\mathbf{c}}_a$ | Anchor center | $\hat{I}^{(n)}$ | Estimated projection |
| $\mathbf{s}_i$ | Gaussian scale | $\bar{\boldsymbol{f}}_a$ | Anchor feature | $\mathbf{z}^{(n)}$ | Particle latent code |
| $\Delta\boldsymbol{c}_i$ | Gaussian center offset | $\bar{\boldsymbol{R}}_a$ | Anchor rotation | $\mathbf{R}^{(n)}$ | Particle rotation |
| $\Delta\boldsymbol{f}_i$ | Gaussian feature offset | $\bar{\boldsymbol{t}}_a$ | Anchor translation | $\mathbf{t}^{(n)}$ | Particle translation |
| $\boldsymbol{t}_i^{(n)}$ | Gaussian translation | | | | |

Table 1: Summary of notations used to denote variables related to Gaussians, anchors or particles.

anchors as, $\mathcal{A} = \{(\bar{c}_a, \bar{\boldsymbol{f}}_a)\}_{a=1}^{A}$, where $\bar{c}_a \in \mathbb{R}^3$ specifies the anchor center location in a canonical frame, and $\bar{\boldsymbol{f}}_a \in \mathfrak{F} \subset \mathbb{R}^E$ is an associated feature vector that encodes heterogeneity information of its corresponding part. The GMM has $G$ Gaussian components associated with anchors (Fig. 2B, left), denoted by $\mathcal{G} = \{(\boldsymbol{f}_i, \boldsymbol{c}_i, s_i, m_i, a_i)\}_{i=1}^{G}$ where $\boldsymbol{f}_i \in \mathfrak{F}$ and $a_i \in \{1, \dots, A\}$ specifies the anchor associated with the Gaussian that is set by the part discovery method below.

We parameterize the position and feature embedding of the $i$-th Gaussian relative to its associated anchor $a_i$ as

$$\boldsymbol{c}_i = \bar{\boldsymbol{c}}_{a_i} + \Delta\boldsymbol{c}_i \,, \quad \boldsymbol{f}_i = \bar{\boldsymbol{f}}_{a_i} + \Delta\boldsymbol{f}_i \,, \tag{3}$$

where $\Delta\boldsymbol{c}_i \in \mathbb{R}^3$ and $\Delta\boldsymbol{f}_i \in \mathbb{R}^E$ are learnable offsets. We initially set $\Delta\boldsymbol{f}_i = \boldsymbol{0}$ so all Gaussians are initialized with the features of their corresponding anchors.

To enable conformational variability, we parameterize deformations at two levels. First, the large-scale motion of each anchor frame is parameterized as a rigid body transformation (Fig. 2C, top). Given the latent code for $n$-th particle image, $\boldsymbol{z}^{(n)} \in \mathfrak{Z}$, and the anchor feature vector $\bar{\boldsymbol{f}}_{a_i}$, we compute the rotated and translated center of the $i$-th Gaussian, $\tilde{\boldsymbol{c}}_i^{(n)}$, as

$$\tilde{\boldsymbol{c}}_i^{(n)} = \bar{\boldsymbol{R}}_{a_i}^{(n)}\Delta\boldsymbol{c}_i + (\bar{\boldsymbol{c}}_{a_i} + \bar{\boldsymbol{t}}_{a_i}^{(n)}) \,, \quad \text{where } \bar{\boldsymbol{R}}_{a_i}^{(n)}, \bar{\boldsymbol{t}}_{a_i}^{(n)} = \text{MLP}^{\mathcal{A}}([\bar{\boldsymbol{f}}_{a_i}, \boldsymbol{z}^{(n)}]; W^{\mathcal{A}}) \,, \tag{4}$$

where $[\cdot, \cdot]$ denotes concatenation, and the MLP with weights $W^{\mathcal{A}}$ returns a rotation $\boldsymbol{R}_{a_i}^{(n)} \in SO(3)$ and translation vector $\boldsymbol{t}_{a_i}^{(n)} \in \mathbb{R}^3$. To capture fine-scale flexibility, additional shifts are applied to individual Gaussians (Fig. 2C, middle), i.e.,

$$\boldsymbol{c}_i^{(n)} = \tilde{\boldsymbol{c}}_i^{(n)} + \boldsymbol{t}_i^{(n)} \,, \quad \text{where } \boldsymbol{t}_i^{(n)} = \text{MLP}_{\boldsymbol{c}}^{\mathcal{G}}([\boldsymbol{f}_i, \boldsymbol{z}^{(n)}]; W_{\boldsymbol{c}}^{\mathcal{G}}) \,. \tag{5}$$

Here, the network $\text{MLP}_{\boldsymbol{c}}^{\mathcal{G}}$, with separate weights $W_{\boldsymbol{c}}^{\mathcal{G}}$, generates individual Gaussian displacements, $\boldsymbol{t}_i^{(n)} \in \mathbb{R}^3$, which are smooth as Gaussians associated with the same anchor will have similar features.

Finally, to account for compositional variability, where regions of a density map may be missing, we modulate Gaussian amplitudes (Fig. 2C, bottom), as

$$m_i^{(n)} = m_i \times \sigma_i^{(n)} , \quad \text{where } \sigma_i^{(n)} = \text{MLP}_m^{\mathcal{G}}([\boldsymbol{f}_i, \boldsymbol{z}^{(n)}]; W_m^{\mathcal{G}}) . \tag{6}$$

Here, $\text{MLP}_m^{\mathcal{G}}$ is an MLP with a sigmoid output activation to restrict the modulation to $(0, 1)$. Values close to $0$ and $1$, respectively, correspond to inactive (absent) and active (present) Gaussians. Considering both modifications to centers and amplitudes, we obtain a modulated set of 3D Gaussians for $n$-th particle image, $\mathcal{G}^{(n)} = \{(\boldsymbol{c}_i^{(n)}, s_i, m_i^{(n)})\}$. Gaussian scales remain the same as they control local resolution, a factor independent of structural variability.

We jointly optimize the parameters $\Theta$ (which includes Gaussian and anchor parameters and MLP weights), and the per-image latent coordinates, $Z = \{z^{(n)}\}$, by minimizing the objective (Fig. 2D)

$$L(\Theta, Z) \;=\; \frac{1}{N} \sum_{n=1}^{N} \mathcal{L}\left(I^{(n)}, \hat{I}^{(n)}\right) \;+\; \lambda_z \, ||\boldsymbol{z}^{(n)}||_2^2 \;+\; \lambda_f \sum_{i=1}^{G} ||\Delta \boldsymbol{f}_i||_2^2 \;, \tag{7}$$

where the reconstruction loss, $\mathcal{L}$, is proportional to the negative image log-likelihood (i.e., the squared error between $I^{(n)}$ and $g^{(n)} \star \hat{I}^{(n)}$ where $g^{(n)}$ is the microscope point spread function and $\hat{I}^{(n)}$ is the 2D projection of $\mathcal{G}^{(n)}$ from Eq. 2). The second term imposes a zero-mean Gaussian prior over the per-image latent codes, ensuring latent coordinates remain bounded [26, 33], while the third term regularizes Gaussians to remain close to the anchor in the feature space. $\lambda_z$ and $\lambda_f$ are hyperparameters that control the relative strength of these priors.

## 3.2 Part Discovery for Model Initialization

The part discovery process is illustrated in Fig. 2A. We optimize a coarse-grained model without anchors and with fewer Gaussians, similarly parameterized as $\mathcal{G} = \{(\boldsymbol{f}_i, \boldsymbol{c}_i, s_i, m_i)\}_{i=1}^{G}$. Here, the Gaussian features, $\boldsymbol{f}_i$, are directly learnable parameters (and randomly initialized). We use $\text{MLP}_c^{\mathcal{G}}$ (Eq. 5), to shift Gaussian centers and $\text{MLP}_m^{\mathcal{G}}$ (Eq. 6) to modulate Gaussian amplitudes. The parameters are estimated using the L2 reconstruction loss and the latent prior, similar to the objective in Eq. 7. Once optimized, we find that the feature space naturally groups Gaussians into 3D parts that undergo consistent motion or appear and disappear together. Remarkably, this property emerges without any direct supervision on features.

To obtain parts, we apply clustering on the Gaussian features, thereby finding regions with reasonably consistent motion and presence. We then further divide these clusters by clustering in 3D space to ensure large parts are well-covered with anchors. For clustering we simply use k-means++ [1]. We use the position and feature vector of the Gaussian closest to the centroid of the cluster to initialize the anchor set, $\mathcal{A} = \{(\bar{\boldsymbol{c}}_a, \bar{\boldsymbol{f}}_a)\}_{a=1}^{A}$. From the coarse-grained model, we also compute an improved density map which is used to seed the Gaussians of the part-aware model. This provides a more robust initialization, especially in the presence of large-scale motion which can lead to blurred or over-dispersed density. Lastly, the coarse-grained model provides a preliminary estimate of the image latent codes, which are used to initialize latent codes in the part-aware model.

**Remark.** Methods for 4D scene reconstruction [27, 31], and DynaMight [40] in cryo-EM, often use neural networks to output deformations or motion. However, they condition on positional encodings of input coordinates instead of learnable features. Such fixed conditioning strongly biases deformations to be spatially smooth, whereas our approach with learnable feature space enables a more flexible form of piecewise smoothness, allowing nearby parts to move quite differently. Through an ablation study, we show that positional encodings quantitatively underperform as well.

## 4 Experimental Setup

We quantitatively compare CryoSPIRE with the state-of-the-art methods, namely, RECOVAR [10], CryoDRGN [47], DRGN-AI [19], 3DFlex [33] and 3DVA [32] using the CryoBench benchmark [15]. We also provide qualitative results on experimental datasets.

**CryoBench.** The sole benchmark for cryo-EM heterogeneity is CryoBench [15], a set of synthetic datasets with ground-truth labels and a protocol for quantitative evaluation. Two datasets, *IgG-1D*

| Method | IgG-1D | | IgG-RL | | Ribosembly | |
|---|---|---|---|---|---|---|
| | Mean (std) | Med | Mean (std) | Med | Mean (std) | Med |
| 3D Classification [39] | 0.297 (0.019) | 0.291 | 0.309 (0.01) | 0.307 | 0.289 (0.081) | 0.288 |
| CryoDRGN [47] | 0.366 (0.003) | 0.366 | 0.349 (0.008) | 0.348 | 0.415 (0.019) | 0.415 |
| CryoDRGN-AI-fixed [19] | 0.366 (0.001) | 0.366 | 0.355 (0.007) | 0.354 | 0.372 (0.032) | 0.374 |
| 3DFlex [33] | 0.336 (0.002) | 0.336 | 0.339 (0.007) | 0.339 | - | - |
| 3DVA [32] | 0.351 (0.003) | 0.351 | 0.341 (0.006) | 0.341 | 0.375 (0.038) | 0.372 |
| RECOVAR [10] | 0.391 (0.001) | 0.391 | 0.372 (0.008) | 0.371 | **0.430 (0.016)** | **0.432** |
| CryoSPIRE (ours) | **0.402 (0.002)** | **0.402** | **0.386 (0.014)** | **0.389** | 0.427 (0.014) | 0.424 |

Table 2: Mean (standard deviation) and median of AUC of Per-Conformation FSCs on CryoBench datasets [15]. Statistics computed across different structural states, i.e. 100 for IgG-1D and IgG-RL and 16 for Ribosembly (Best method in bold, second best underlined).

and *IgG-RL*, are based on the human immunoglobulin G (IgG) complex, simulating conformational changes by rotating the dihedral angle between the Fab domain and the IgG core (see Fig. 4D), generating 100 distinct conformations, each with 1,000 particle images. *Ribosembly* simulates compositional heterogeneity by successively adding protein subunits and ribosomal RNA, resulting in 16 discrete structural states [35]. It has 335,240 particle images, with non-uniform distribution over the 16 compositional states. All particle images have $128 \times 128$ pixels, and are simulated with realistic point spread functions and a signal-to-noise ratio (SNR) of 0.01.

**Experimental Datasets.** We also evaluate on two real datasets: EMPIAR-10076 is a dataset comprising assemblies of intermediates of the *Escherichia coli* large ribosomal subunit (LSU) [7], with 131,899 particle images ($320 \times 320$ pixels, with pixel size 1.31 Å). In the original study, four major assembly states were identified [7], with a subset of particles labeled as unassigned (non-ribosomal impurities) or 30S subunits. We also consider EMPIAR-10180, a conformationally heterogeneous dataset of Pre-Catalytic Spliceosome [30]. A total of 327,490 particle images were collected ($320 \times 320$ pixels, with pixel size 1.69 Å). Consistent with other heterogeneity methods considering this dataset [10, 47] we perform analysis on a filtered subset of 139,722 images.

**Implementation Details.** For part discovery, we seed $G = 2,048$ components using the rigid reconstruction and adopt lightweight MLPs with a single hidden layer of $H = 32$ units. The latent space, $\mathfrak{Z}$, has dimensionality $D = 4$ and the feature space, $\mathfrak{F}$, has dimensionality $E = 24$. We optimize the part discovery model for 15 and 50 epochs on synthetic and experimental datasets. The part-aware GMMs are optimized for 30 epochs, using $G = 8,192$ components, except for Ribosome synthetic and experimental datasets with $G = 16,384$, and have MLPs with three hidden layers and $H = 128$ hidden units. On experimental datasets, we perform part discovery on downsampled $128 \times 128$ images for efficiency, while the part-aware GMM is optimized on $256 \times 256$ images. We use batch size $B = 64$ and set hyperparameters $\lambda_z = 0.1$, $\lambda_f = 0.01$. The optimization runs on a single NVIDIA GeForce RTX 2080, taking 3 to 6 hours depending on the number of Gaussians in the model.

**Evaluation Metrics.** The quality of cryo-EM density maps are evaluated using Fourier Shell Correlation (FSC) [43], which is the normalized cross-correlation between two independently estimated density maps, as a function of frequency. Metrics for heterogeneity are less standardized, but the most common is Per-Conformation FSC (or Per-Conf FSC) [15], proposed by CryoBench [15]. Per-Conf FSC is the average FSC between the ground-truth 3D structure of a particle state, and the 3D structure corresponding to the average latent position of images associated with that state. The Per-Conf FSC requires knowledge of ground-truth 3D structures for each image which is not available for experimental data and we instead rely on qualitative evaluation of the estimated parts and structures. FSC results in a curve which can be summarized by computing the area under the curve (AUC) to more easily compare methods. See the supplement for more details on metrics.

## 5 Results

Quantitative comparison on the three relevant CryoBench [15] datasets are provided in Table 2 and Fig. 3. Note that CryoSPIRE outperforms 3DVA and 3DFlex which are among the most widely used methods in cryo-EM at present. As 3DFlex cannot handle compositional changes, it was not evaluated on Ribosembly. CryoSPIRE outperforms non-linear latent variable models, Cryo-DRGN

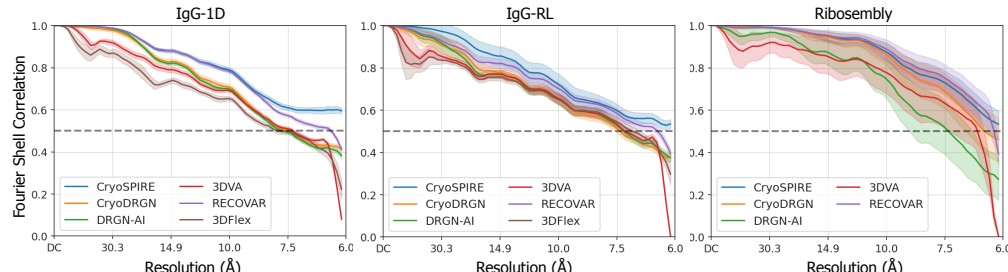

Figure 3: Per-Conformation FSC on CryoBench datasets. Error bars indicate standard deviation across different conformations. The highest possible resolution is 6 Å on these synthetic datasets.

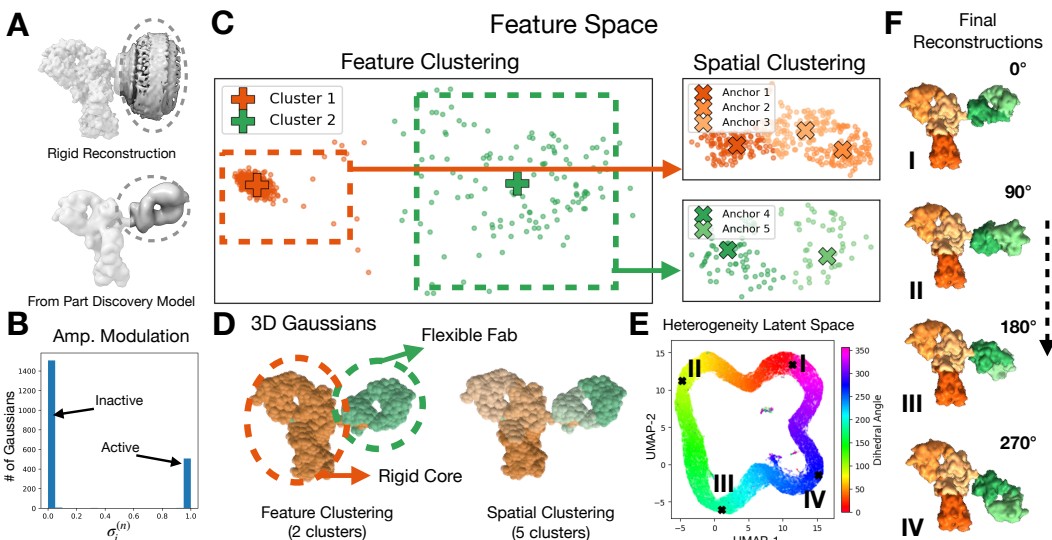

Figure 4: Results on IgG-1D [15]. **(A)** Due to large motion, the Fab domain (circled) is smeared out in rigid reconstruction, while our part discovery model identifies this domain and resolves its structure and motion, providing good initialization for subsequent modeling. **(B)** For a sample structure, the histogram of amplitude modulations indicate active and inactive Gaussians. **(C)** Gaussian feature space, $\mathfrak{F}$, shows two distinct groups (green, orange), corresponding to the flexible Fab and the rigid core; feature clustering finds these groups and divides further based on spatial proximity, yielding 5 parts. **(D)** Configuration of 3D Gaussians after Level-1 and Level-2 clustering. **(E)** The latent space, $\mathfrak{Z}$, captures conformation change (colored based on ground truth Fab orientation). **(F)** Sample structures from model corresponding to four latent points, showing rotation of the Fab domain (green).

and DRGN-AI, especially on IgG-1D and IgG-RL by a large margin. The most competitive method is the linear subspace model of RECOVAR, which, as reported, is memory intensive due to allocation of several bases and is not as interpretable without motion modeling. While CryoSPIRE significantly outperforms RECOVAR on IgG datasets, its performance on Ribosembly, where linear subspace models are more favorable by design, is not statistically different from RECOVAR. Relative to the nominal FSC threshold of 0.5 for comparison to ground truth [36], the FSC curves in Fig. 3 indicate that CryoSPIRE finds higher resolution density maps. Finally, we note that CryoSPIRE is the first GMM-based method to be successfully evaluated on CryoBench.

**IgG-1D & IgG-RL (CryoBench [15]).** The flexible Fab domain (circled in Fig. 4A, top) in the rigid reconstruction, used as input for part discovery, is poorly resolved. However, the part discovery model learns to selectively deactivate incoherent parts, as shown in the histogram of the modulation factors $\sigma_i^{(n)}$ in Fig.4B. This enables a more robust initialization (Fig. 4A, bottom) of the hierarchical GMM. The Gaussian feature space, $\mathfrak{F}$, shows two clusters corresponding to the flexible Fab domain from the rigid core (Fig. 4C for IgG-1D and Fig. 5B for IgG-RL). Spatial clustering produces a total of five and six anchors for IgG-1D and IgG-RL, respectively. The latent heterogeneity space, $\mathfrak{Z}$,

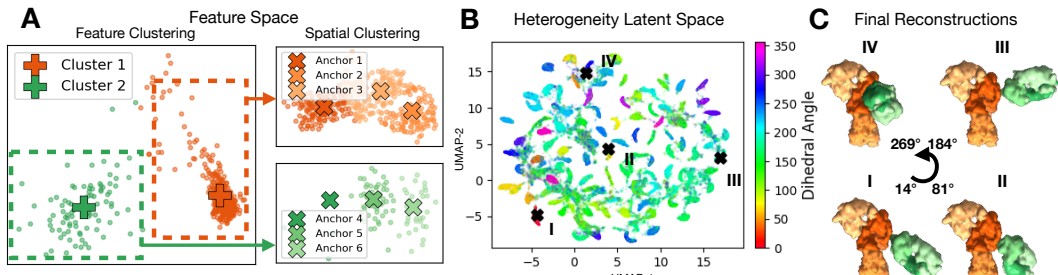

Figure 5: Results on IgG-RL [15]. **(A)** The feature space, $\mathfrak{F}$, shows two parts (green and orange) corresponding to the flexible Fab domain and the rigid core. Subsequent spatial clustering yields six parts. **(B)** The latent space, $\mathfrak{Z}$, is colored with Fab orientation along with four sampled latent points that capture rotation of the Fab domain (comprising three parts). The motion of the Fab domain in IgG-RL is not as regular as that in IgG-1D, as reflected in the latent space. **(C)** The corresponding density maps are provided with parts illustrated in different colors.

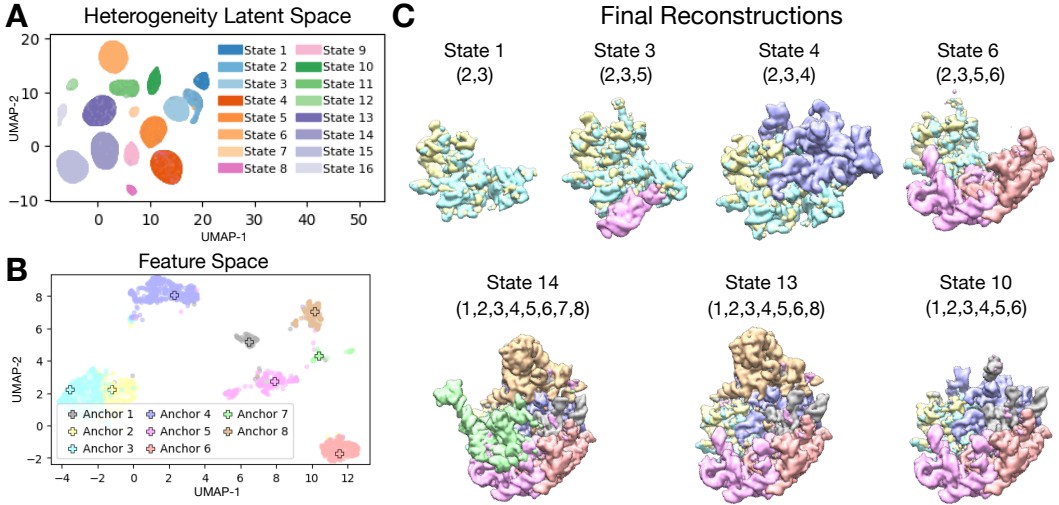

Figure 6: Results on Ribosembly [15] **(A)** Gaussian feature space, $\mathfrak{F}$, showing eight major parts identified through clustering. **(B)** Heterogeneity latent space, $\mathfrak{Z}$, colored coded with the ground-truth compositional state. **(C)** Visualizations of 3D density maps corresponding to seven points in latent space, with colors depicting parts (given in parentheses).

indicates a circular manifold of dihedral angles for IgG-1D, see Fig. 4D. Four structures from the latent space in both datasets demonstrate that the Fab domain, covered by a few parts, undergo a large, predominantly rigid motion, while the rest of the complex remains fixed.

**Ribosembly (CryoBench [15]).** After part discovery, we obtain eight parts (see Fig. 6A) that are used to initialize eight anchors in the part-aware GMM. In Fig. 6B the learned latent space, $\mathfrak{Z}$, clearly distinguishes between the different compositional states. For seven selected states, we visualize the estimated structure (Fig. 6C), colored based on the discovered parts.

**Large Ribosomal Subunit (EMPIAR-10076 [7]).** We find four major assembly states in the part discovery latent space (labeled as (I, II, V, VI) in Fig. 7A, left), which match classes (C, E, B, D) in the original study [7], with unassigned particles and 30S contaminants grouped in states III and IV, which are excluded when optimizing hierarchical model (See supplement for more details). The Gaussian feature space, $\mathfrak{F}$, (Fig. 7B) shows four distinct parts which also align with previously reported structural blocks in the original study (cf. [7], Fig. 6). By analyzing the heterogeneity latent space, $\mathfrak{Z}$, of the part-aware model (Fig. 7A, right), we show that the major states can be further divided into subpopulations; e.g., the major state I is represented with minor states (1, 2) and the major state II has branched into minor states (3, 4, 5). The associated structures, shown in Fig. 7E, are consistent with minor states reported in the original study [7].

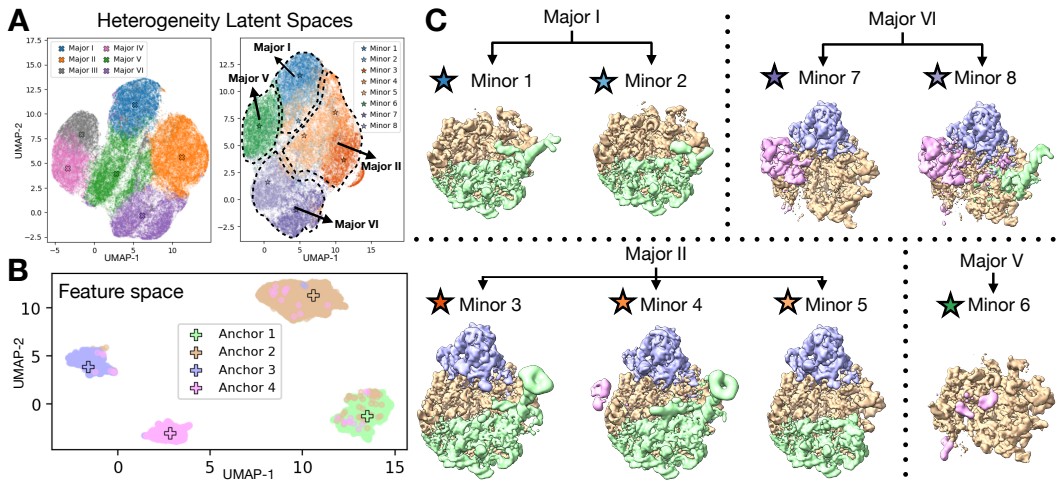

Figure 7: Results on Large Ribosomal Subunit (EMPIAR-10076 [7]). **(A)** The learned heterogenity latent spaces, $\mathfrak{Z}$, in part discovery (left) identifies the four major assembly states (I, II, V, VI) and two groups of impurities (III, IV). After fitting the part-aware model, the major states, with impurities excluded, can be further categorized (right) into eight color-coded minor structural states. **(B)** The part discovery Gaussian feature space, $\mathfrak{F}$, reveals four parts which are used to construct the part-aware model. **(C)** The structures corresponding to different states, colored by inferred part.

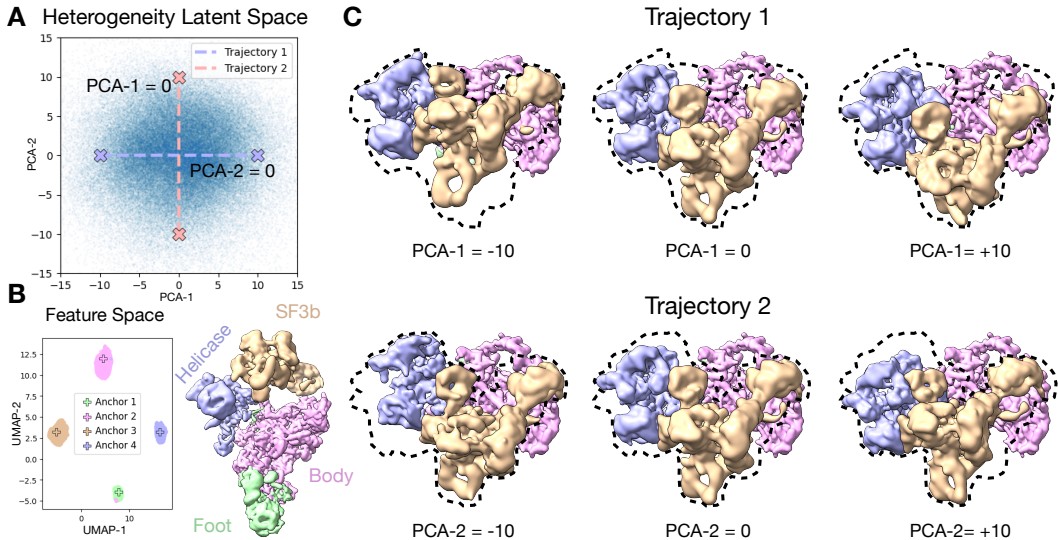

Figure 8: Results on Pre-Catalytic Spliceosome (EMPIAR-10180 [30]). **(A)** PCA of the latent space, $\mathfrak{Z}$, is used to generate two structural trajectories. **(B)** The Gaussian feature space, $\mathfrak{F}$, shows four parts which correspond to known helicase, SF3b, body and foot domains as shown in 3D visualization of Gaussian components configuration. **(C)** Three states along each trajectory. In both trajectories, body is rigid while SF3b and helicase show large-scale motion.

**Pre-Catalytic Spliceosome (EMPIAR-10180 [30]).** The feature space, $\mathfrak{F}$, of the part discovery model (Fig. 8B), shows four distinct clusters, which correspond to coherent structural regions – foot, body, helicase, and SF3b – consistent with the original study [30]. Accordingly, we optimize the part-aware model with four anchors. To illustrate structural variability, we run PCA on the heterogeneity latent space, $\mathfrak{Z}$, and extract two principal directions illustrated in Fig. 8A. Top views of density maps along the two principal directions (Fig. 8C) show two modes of conformational heterogeneity. The first direction reflects a forward–backward rotation of the SF3b and helicase regions. The second direction captures a side-to-side rotation of SF3b, and a diagonal shift of the helicase. Please see the supplement for more visualization on conformational changes.

| Method | IgG-1D | IgG-RL | Ribosembly |
|---|---|---|---|
| CryoSPIRE | **0.402 (0.002)** | **0.386 (0.014)** | **0.427 (0.014)** |
| w/o hier. motion | 0.388 (0.002) | 0.372 (0.010) | 0.425 (0.015) |
| over-segment | 0.384 (0.002) | 0.375 (0.010) | 0.423 (0.016) |
| w/ pos. encoding | 0.377 (0.002) | 0.361 (0.007) | 0.415 (0.023) |

Table 3: Mean AUC-FSCs reported on datasets from CryoBench [15] for ablation study.

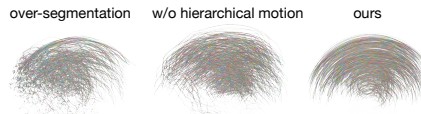

Figure 9: Estimated motion of Gaussians for 30 states of IgG-1D. The baselines fail to capture local rigidity.

## 5.1 Ablation Study

Here, we ablate key design decisions in our framework. To demonstrate the importance of anchor-based motion modeling in CryoSPIRE, we consider a baseline without anchors that uses an MLP to directly learn deformations of individual Gaussians. Quantitative comparison on IgG-1D and IgG-RL, as in Table 3, shows that the lack of anchor based motion leads to inferior results. This is less critical for Ribosembly with minor conformational changes. We also compare with a model that over-segments the structure by using $K = 64$ anchors, which achieves worse performance. In Fig. 9, we visualize the estimated motion of Gaussians on the IgG-1D dataset. Both baselines fail to capture the locally rigid and smooth motion. Finally, we consider a baseline where the Gaussian feature space is replaced with a positional encoding, similar to previous methods, e.g., [40]. This baseline is unable to identify meaningful parts and achieves inferior quantitative performance.

## 6 Conclusion

We present CryoSPIRE, a hierarchical cryo-EM density model to capture conformational and compositional heterogeneity in the 3D structure of biomolecules from 2D images. This includes a novel method for part discovery and a hierarchical Gaussian mixture model for which the parts provide meaningful inductive biases to regularize model fitting. CryoSPIRE establishes a new state-of-the-art on the CryoBench heterogeneous benchmark, and produces meaningful parts on experimental data.

While CryoSPIRE shows promising results, limitations exist. First, validation of estimated structures and variability from heterogeneous experimental data remains an open problem for all methods, including CryoSPIRE. Second, interpreting the inferred latent space remains challenging, specifically how it may relate to the biophysical energy landscape of molecular states. Third, learning per-Gaussian features is a key design choice in cryoSPIRE, as it provides the inductive bias that drives features to encode local structural heterogeneity. To that end, we have only used very simple algorithms like k-means++, which requires manual selection of the number of clusters (parts). Further research will be useful to find more effective forms of clustering, perhaps incorporating principled biophysics criteria like free energy. Finally, like other methods, we presume an initial estimate of the structure and image poses; inaccuracies in these may limit CryoSPIRE's efficacy. A fully *ab initio* method for heterogeneous data remains an open problem.

## Broader Impact

Cryo-electron microscopy (cryo-EM) has emerged as a revolutionary technique in structural biology, enabling the determination of macromolecular structures with significant societal impact. Computational methods, grounded in machine learning and computer vision have now been used to determine many thousands of biological structures. Notably, cryo-EM played a pivotal role in elucidating the structure of the SARS-CoV-2 spike protein, revealing its pre-fusion conformation and aiding in the assessment of medical countermeasures. Complementing computational methods such as AlphaFold for protein structure prediction, cryo-EM has revolutionized our understanding of cellular processes and accelerated the development of novel therapeutics, including synthetic antibodies. Nevertheless, we strongly condemn any usage of our proposed hierarchical 3D GMM representation for generating malicious data, improperly modifying signals, or spreading misinformation.

## Acknowledgments and Disclosure of Funding

We thank Ali Punjani and John Rubinstein for helpful discussions. This research was supported in part by the Province of Ontario, the Government of Canada (through NSERC, CIFAR, and the Canada First Research Excellence Fund for the Vision: Science to Applications (VISTA) programme), and by the companies sponsoring the Vector Institute for Artificial Intelligence.

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
