# OpenReview forum: "Reconstructing Heterogeneous Biomolecules via Hierarchical Gaussian Mixtures and Part Discovery"
_NeurIPS.cc/2025/Conference — NeurIPS 2025 poster_

### Official Review · Reviewer_njmf · 2025-06-21

**Clarity:** 2
**Significance:** 3
**Originality:** 2
**Rating:** 4
**Confidence:** 3

**Summary:**

This paper proposes a framework for 3D reconstruction of molecular structure from noisy 2D images using hierarchical Gaussian mixture model. The method first infers a part-based segmentation using a coarse-grained GMM with learnable features. Rotations, translations and amplitude modulations are introduced to account for conformational flexibility and compositional variation. Numerical experiments on multiple datasets show the efficiency and utility of the proposed framework comparing to benchmarks.

**Questions:**

1. Could you please clarify the connection and difference between Eq (1)-(2) in the main paper and Eq (8)-(10) in the supplement? Do you assume ${\bf R}^{(n)}$ and ${\bf t}^{(n)}$ are known or unknown? They do not appear in Eq (8)-(10).
2. Could you please clarify how do you specify the number of anchors $A$, number of Gaussians $G$ and which Gaussian is associated with which anchor? Would these affect the results of reconstruction?
3. Could you please comment on the convergence performance of reconstruction loss minimization?
4. I noticed there are a lot of uncertainties in FSC metric for Ribosembly dataset for all methods but much fewer for the other two datasets (Fig 3). Could you please comment on this? Where do you think the difference comes from?

**Ethical Concerns:**

["NO or VERY MINOR ethics concerns only"]

**Final Justification:**

Dear AC, SAC, PC:

I think the author partly addressed my concerns regarding sensitivity and robustness in their latest rebuttal. But since most of my training is in statistics and not in biology, I may not be able to appreciate the practical significance of the work. However, this work seems to have significant practical contributions according to other reviewers. I feel the writing clarity can be further improved, but this does not seem to be a concern to others as well.
In the end, I decided to raise my score by 1 and keep my confidence. I hope this can help with the final decision. Thank you!

Best,
Reviewer njmf

**Limitations:**

The connection between Section 2 and Section 3 is not very clear. The new notation in Section 3 Eq(3)-(5) is confusing--it may be better to reconsider the notation or provide a summary in a table.

**Paper Formatting Concerns:**

None.

**Quality:**

2

**Strengths And Weaknesses:**

Strengths:

1. The authors introduce the practical problem and limitations of existing methods very well, which provides a clear motivation of the proposed method and its significance.
2. The framework is very flexible and fits nicely with practical considerations.

Weaknesses:

1. Notation is confusing and writing clarity can be improved.
2. Due to the complexity of the model, the identifiability of parameters and the robustness of the results are unclear.

---

> ### Author Rebuttal · Authors · 2025-07-30
>
> We thank the reviewer for finding our framework flexible and practical. We will address the reviewer’s comments and questions below.
>
> **"Notation is confusing and writing clarity can be improved ... The connection between Section 2 and Section 3 is not very clear. The new notation in Section 3 Eq(3)-(5) is confusing--it may be better to reconsider the notation or provide a summary in a table."**
> We will revise the paper to clarify these sections, draw clear connections and unify notations. In short, Sec. 2, presents background on Gaussian mixture models (GMM), and it provides the well-known rendering equation (Eq. 2) to map a 3D GMM onto the image plane, with notations that makes explicit the amplitude ($m_i$), scale ($s_i$) and center ($c_i$) parameters for the i-th component. Sec. 3 presents our method, using the same notation to denote the GMM parameters. But we add a particle index $n$ in the superscript to allow us to have per-particle structural variation. The structural variation is captured by the modulation of amplitudes and the displacement of Gaussian centers (see Eqs 4-6). Following your suggestion, we will add a table that summarizes notation and rewrite these sections for improved clarity.
>
> **"Due to the complexity of the model, the identifiability of parameters and the robustness of the results are unclear."**
> Cryo-EM (with or without heterogeneity) is a highly non-convex problem with multiple minima. As is the case with GMM fitting, there are inherent symmetries where, for example, permutations of the parameters yield similar or even identical solutions. As such, like learning in neural networks, one should assume that the parameters are not straightforwardly identifiable, and robustness is hard to establish in any formal sense. Note that, in terms of robustness, since the introduction of SGD for ab-initio by cryoSPARC in 2017, single-particle rigid cryo-EM reconstruction has been generally successful in practice. Methods for heterogeneity, while not as reliable as methods for the single, rigid particle case, have been effective and continue to improve.
>
> **"Could you please clarify the connection and difference between Eq (1)-(2) in the main paper and Eq (8)-(10) in the supplement? Do you assume R(n) and t(n) are known or unknown? They do not appear in Eq (8)-(10)."**
> We apologize for the confusion. In all experiments, we assume that pre-estimated particle poses (parameterized by rotation matrix $R$ and translation vector $t$) are given as input. For Eqs 8-10 in the supplement, to avoid clutter, we dropped the dependence on $R$ and $t$, assuming the particles are already reoriented and positioned based on the input pose. For clarity, we will modify the supplement to make this explicit and avoid such confusion.
>
> **"Could you please clarify how do you specify the number of anchors A, number of Gaussians G and which Gaussian is associated with which anchor? Would these affect the results of reconstruction?"**
> To determine the number of anchors/parts, following the optimization of the part discovery GMM, as discussed in L162-170, we perform dimensionality reduction with UMAP to visualize the learned Gaussian feature space. As illustrated in Figs. 4C, 5A, 6B, 7B, and 8B, Gaussian components are clustered into well-separated groups, indicating groups of Gaussians with similar properties. We then select the desired number of parts (i.e. clusters) and use kmeans++ to obtain assignment of Gaussians to parts. Our results show a great promise for future work on automatic methods to find the number of clusters.
>
> While the part discovery stage uses a small number of Gaussian components (i.e. 2048), the hierarchical model in the second stage captures more fine-grained details, using many more Gaussians (8192 or 16384 based on the structure size). To assign Gaussians to anchors (part centers so to speak), we take the following procedure: For each Gaussian in the hierarchical model, we find the closest Gaussian in the part-discovery model, and use the part to which it’s assigned. Please see Sec. D in the supplement for more thorough discussion of this procedure.
>
> Finally, in an ablation study (Sec. 5.1) we show that using a high number of anchors (eg, $A=64$) leads to significant oversegmentation and degrades performance (see Table 2), as it fails to capture locally rigid and smooth motion (illustrated in Fig. 9, left). While there may be no ideal number of anchors from first principles, we find that 5-10 anchors are sufficient to help provide useful priors for heterogeneity in the datasets that we’ve considered.
>
> **"Could you please comment on the convergence performance of reconstruction loss minimization?"**
> In cryoEM methods, convergence is generally assessed in terms of the stabilization of the 3D reconstruction.  Like other non-convex optimization problems in machine learning, there are few principled ways to assess convergence.  In CryoEM, because of the high level of image noise, the loss and its gradients are also very noisy. As a consequence, relatively large batches are typically used to reduce the variance of the gradient estimator during optimization, and the minibatch loss is not a very reliable indicator of convergence.  We find 30 epochs are sufficient for the map to stabilize.
>
> **"I noticed there are a lot of uncertainties in FSC metric for Ribosembly dataset for all methods but much fewer for the other two datasets (Fig 3). Could you please comment on this? Where do you think the difference comes from?"**
> The visualization in Fig. 3 shows the FSC curve averaged over all structural states. The error bars depict the FSC standard deviation across different states. There are two likely causes for the FSC variability across structural states:  First, with Ribosembly, unlike IgG, particles are not uniformly distributed across compositional states. States with few particles are expected to be determined at lower resolutions (i.e. lower FSC curves) than those with many particles. A second cause may be differences in the type of heterogeneity. With conformational variability, as in IgG, if one can accurately model deformation, then one can aggregate information across many particles to resolve structure more accurately, yielding consistent FSC curves.  With compositional heterogeneity, parts of density be present in only a small number of states (e.g., consider a small subunit present in only one state) and as a result FSCs may vary significantly for different states, producing greater variance.

---

> ### Comment · Reviewer_njmf · 2025-08-01
>
> Thank you for the rebuttal! I have decided to keep the score due to concerns regarding the parameter choices, convergence and robustness of the algorithm.

---

> > ### Author Response · Authors · 2025-08-02
> > **Robustness**
> >
> > Thank you for the prompt response! While we tried to provide clarifications in our rebuttal, the concerns in the review were not very specific. If there is a specific issue concerning robustness and convergence, or an experiment you are looking for, please let us know, and we will be able to address it.
> >
> > We reiterate that robustness and convergence are important matters in cryoEM as a high dimensional nonconvex problem with high noise levels. In our experience, our method is robust, both on the datasets in the paper as well as other ones we have explored since the submission. We have run cryoSPIRE method multiple times with different random seeds and see consistent, SOTA results on each run. We use the same hyper-parameter settings across all datasets, except for the number of parts and Gaussians, which scale naturally with the size and complexity of the protein and are clearly evident using standard visualizations of the features.

---

> > > ### Comment · Reviewer_njmf · 2025-08-04
> > >
> > > Dear authors,
> > >
> > > Thanks for the response! I raised my concerns about evaluation of model convergence and robustness because I would like to see
> > >
> > > 1. Whether there are theoretical results that guarantee convergence of the proposed method, and how good the converged point is. You mentioned "convergence is generally assessed in terms of the stabilization of the 3D reconstruction", but I think this is rather qualitative than quantitative, and I am not convinced if this is sufficient or reliable.
> > >
> > > 2. Sensitivity analysis, how modeling choices you made affect the results:
> > >
> > > 2.1 For the number of parts (anchors), you used UMAP to visualize in 2D and decide number of clusters, but this is again a bit ad hoc to me, especially in complicated cases like Figure 6. What if you choose a much smaller/larger number of parts?
> > >
> > > 2.1 You mentioned "5-10 anchors are sufficient to help provide useful priors for heterogeneity in the datasets that we’ve considered." But I believe this number would depend on different datasets, and it is unclear to me how you would provide general guidance for this choice to non-experts.
> > >
> > > Basically it is fine if the model and algorithm breaks under some conditions, but I'd like to know how it can break and how do I know it breaks, especially if in reality I may not know what results I'd expect.
> > > (I'm not an expert in biology, so I'd like to know in reality, whether biologists can know if the reconstruction results they get are good or bad.)

---

> > > > ### Author Response · Authors · 2025-08-07
> > > >
> > > > We thank the reviewer for the clarification of their concerns.
> > > >
> > > > **Convergence.** CryoEM reconstruction methods aim to solve non-convex optimization problems, even in the relatively simpler setting of homogenous structure estimation due to the unknown poses. As a result, and like in training deep neural networks, there are generally no theoretical guarantees of global convergence. Our method uses stochastic gradient descent (SGD) which comes with the usual convergence guarantees to local minima under standard assumptions (e.g., smoothness, learning rates, etc).
> > > >
> > > > In practice, convergence monitoring is limited to standard heuristics like the change in loss function and the change in the model parameter space. As noted in our previous comment, the high level of noise in the images limits the usefulness of loss function monitoring directly. Instead we monitor changes in GMM parameters (centers, amplitudes and bandwidths) and run SGD until the magnitude of changes in parameters is relatively small to find a sufficient number of epochs for convergence.
> > > >
> > > > **Sensitivity to the number of parts.** We included a limited ablation of the number of parts in Sec. 5.1 with the extremes of one part and 64 parts (reported as w/ pos. encoding and oversegmentation). To better answer your concern, we have run the Ribosembly dataset with 6 to 10 parts and we summarize the findings in the table below, clearly showing that performance is stable and that our method is not overly sensitive to the number of parts. We will include this more detailed ablation in the revised paper.
> > > >
> > > > |# of Parts|Per-Conf FSC-AUC (mean ± std)|
> > > > |-|-|
> > > > |1 (in paper as w/ pos. encoding)|0.415 ± 0.023|
> > > > |6|0.426 ± 0.015|
> > > > |7|0.427 ± 0.014|
> > > > |8 (in paper)|0.427 ± 0.014|
> > > > |9|0.427 ± 0.015|
> > > > |10|0.426 ± 0.015|
> > > > |64 (in paper as oversegmentation)|0.423 ± 0.016|
> > > >
> > > > **General guidance on the number of parts.** The above ablation shows that the method is relatively robust to the number of parts. Our visualizations of the feature space (e.g. Figs 6B,7B,8B) also provide strong and relatively intuitive guidance on a reasonable range of part numbers. Further, we built a 3D visualization tool using Polyscope python library that allows users to inspect the part-discovery GMM and interactively run k-means++ to find and visualize clustering results with different numbers of parts. This tool helps provide a 3D interpretation of the selected parts and its code will also be released upon acceptance. Taken together, this implies that when considering resolution, it is very plausible for a non-expert to select a number of parts that will achieve good results.
> > > >
> > > > Note that experimentalists routinely set these kinds of parameters in cryo-EM data processing pipelines. See the case studies for CryoSPARC (please look up Data Processing Tutorials in CryoSPARC Guide webpage), the leading software in cryo-EM, which involve several steps with manual setting of parameters. For example, one needs to select the number of clusters to classify and align 2D images (2D classification step), then select 2D classes for further processing of particles; the 3D classification step requires manually selecting the number of 3D structures to be estimated. Biologists are generally comfortable with such parameters and, in fact, often prefer the flexibility of manual setting over automatic tuning.
> > > >
> > > > **How can biologists know if the results are good or bad**. “Gold-standard FSC” is the main protocol in cryo-EM to validate homogenous reconstruction, where a dataset is split in half, two independent reconstructions are performed and the corresponding estimated structures are compared. Unfortunately, as noted in our response to Reviewer iKnc (see “Evaluation Metrics and Datasets”), this procedure is not possible for most complex forms of heterogeneity and may not be accurate for methods with strong structural priors which violates assumptions of FSC. This is a major open problem for all heterogeneity methods and one of the reasons that CryoBench was built with synthetic data to enable rigorous quantitative evaluations.
> > > >
> > > > As a result, for experimental data, evaluation is necessarily somewhat qualitative and may rely on other information. For instance, visibility of secondary structure and other fine details provides information on resolution, certain configurations of amino acids are more plausible (or impossible) due to biochemistry, or a macromolecule may be composed of several subcomplexes which have rough known structures (and may help guide the number of parts). Ground truth is never really available in real cryo-EM experiments and, as such, qualitative evaluations and interpretations play a central role in experimental cryo-EM practice.
> > > >
> > > > In short, while CryoSPIRE, like other tools for heterogeneity in cryo-EM (e.g., cryoDRGN, etc), requires some amount of qualitative assessment when used on real experimental data, it is well within the norms of cryo-EM data processing.

---

> > > > > ### Comment · Reviewer_njmf · 2025-08-08
> > > > >
> > > > > Dear authors,
> > > > >
> > > > > Thanks for the detailed explanation about convergence and sensitivity, it addresses most of my previous concerns. I decided to raise my score.

---

### Official Review · Reviewer_LVtR · 2025-07-01

**Clarity:** 4
**Significance:** 4
**Originality:** 3
**Rating:** 6
**Confidence:** 4

**Summary:**

Single particle cryo electron-microscopy (cryo-EM) is a field whose goal is to reconstruct 3D volumetric maps of molecules such as proteins and ribosomes from large samples of images taken with a transmission electron microscope. A key challenge in this field is heterogeneity within the sample. This includes discrete heterogeneity due to attachment and detatchment of components as well as continuous heterogeneity due to the motion of parts of the molecule.  This paper proposes a new system for 3D reconstruction in the presence of either discrete or continuous heterogeneity that is based on modeling the atomic structure of the molecule as a Gaussian mixture model with rigidly-moving components. A neural network that takes latent parameters is used to parameterize the specific configuration of the molecule in each sampled image and the model is fit to the data using SGD.

**Questions:**

1. Why did you not include the code in this submission?
2. In page 4, line 133 you explain that the MLP returns a rotation in SO(3). I'm assuming that it is given as a 3x3 matrix. How do you enforce it to be a member of SO(3)?
3. In page 9, line 268, you reference Table 1 but did you mean to reference Table 2?

**Ethical Concerns:**

["NO or VERY MINOR ethics concerns only"]

**Final Justification:**

None of the other reviewers raised any major concerns so I maintain that this is an excellent work. I think the authors did a good job in their rebuttals, including mine.

Since this is clearly high-quality work in an important and timely field, I strongly support acceptance provided that the authors will release their code.

**Limitations:**

yes

**Quality:**

4

**Strengths And Weaknesses:**

The method is clearly motivated and explained. It is tested on CryoBench (currently the leading benchmark in this field) where it shows SOTA results.  The paper is very well written and the included main.html sets a very high bar in scientific visualization.  Overall this is a very strong and well-polished contribution.

Weaknesses:
1. The paper does not include running time benchmarks. The authors should include at least one table of runtimes (or plot), as well as memory requirements, alongside the description of the computer/s used to run the benchmarks: CPU, GPU, how many processes, how many threads, etc.
2. For some reason, the authors did not include the code for this review but have promised to release it upon acceptance. If they follow through on their promise then this is a strong accept.

Minor comments:
1. Page 2: "To out knowledge, this is first GMM-based..." -> "this is the first".
2. Page 2: "Such methods are often notoriously memory intensive and have limited expressiveness". I'm not sure what you mean by "limited expressiveness". Methods based on eigenvectors of the covariance matrix can reconstruct any volume, they may just require a large number of eigenvectors to do so.
3. Page 7, line 240: "rest of the complex is remains fixed" -> remove the word "is".

---

> ### Author Rebuttal · Authors · 2025-07-30
>
> We are grateful that the reviewer finds CryoSPIRE to be a strong and well-polished contribution. Below we answer the questions and discuss concerns raised in the review.
>
> **"The paper does not include running time benchmarks. The authors should include at least one table of runtimes (or plot), as well as memory requirements, alongside the description of the computer/s used to run the benchmarks: CPU, GPU, how many processes, how many threads, etc."**
> We thank the reviewer for pointing this out.  We will include a table with these parameters, both for our cryoSPIRE method as well as the other baseline methods on the benchmark data, where we report total runtime, runtime/epoch, number of epochs, and max GPU memory. (3DFlex does not run on Ribosembly). The GPU is a single NVIDIA GeForce RTX 2080. The CPU is Intel(R) Xeon(R) Silver 4110. For our method we include the part discovery stage too (hence 15 + 30 epochs). Note cryoSPIRE runs significantly faster than competitive methods of CryoDRGN, DRGNAI, and 3DFlex. We will include these tables in the revised paper.
>
> IgG-1D and IgG-RL:
> Method|Total Runtime|Max GPU Mem.|Runtime per Epoch|# Epochs
> -|-|-|-|-
> CryoSPIRE|63 min|5.10 GB|1.40 min|45
> CryoDRGN|304 min|1.34 GB|15.20 min|20
> DRGN-AI|443 min|5.55 GB|4.43 min|100
> 3DVA|37 min|0.75 GB|1.85 min|20
> 3DFlex|725 min|2.12 GB|-|-
>
> Ribosembly:
> Method|Total Runtime|Max GPU Mem.|Runtime per Epoch|# Epochs
> -|-|-|-|-
> CryoSPIRE|405 min|10.60 GB|9.00 min|45
> CryoDRGN|2209 min|1.36 GB|44.18 min|50
> DRGN-AI|1679 min|5.57 GB|16.79 min|100
> 3DVA|120 min|0.75 GB|6.00 min|20
>
> **“For some reason, the authors did not include the code for this review but have promised to release it upon acceptance."**
> We will definitely release the code upon acceptance. The method has been implemented in a publicly available framework, so code release will be straightforward.
>
> **"Page 2: Such methods are often notoriously memory intensive and have limited expressiveness". I'm not sure what you mean by "limited expressiveness". Methods based on eigenvectors of the covariance matrix can reconstruct any volume, they may just require a large number of eigenvectors to do so."**
> We agree that by increasing the number of principal components (e.g., as in RECOVAR), any motion could eventually be represented to a given accuracy, but continuous motions, especially large-scale ones, would require a large number of principal components.  In practice this comes at the expense of much higher (often prohibitive) memory consumption and a higher dimensional latent configuration space. We will clarify this in the revised paper.
>
> **"Page 2: "To our knowledge, this is first GMM-based..." -> "this is the first".
> Page 7, line 240: "rest of the complex is remains fixed" -> remove the word "is".
> Page 9, line 268, you reference Table 1 but did you mean to reference Table 2?"**
> Thank you for bringing the typos to our attention. They will be corrected in the revised paper.
>
> **"In page 4, line 133 you explain that the MLP returns a rotation in SO(3). I'm assuming that it is given as a 3x3 matrix. How do you enforce it to be a member of SO(3)?"**
> We follow prior work [1,2] and use the so-called S2S2 6D representation from which we compute a rotation matrix $R$ using Gram-Schmidt, thereby ensuring $R$ is a member of SO(3). In particular, we set the MLP to return a 6D vector and split it into two 3D vectors. We then normalize each to unit length and denote them by $v_1, v_2 \in \mathbb{R}^{3}$ . Next, we compute their cross product and normalize, ie, $v_3 = \frac{v_1 \times v_2}{||v_1 \times v_2||}$, yielding a third unit vector. Note that $v_3$ is orthogonal to $v_1$. By selecting $v_1$ and $v_3$ as the first and third columns of the rotation matrix, we compute the second column as $\tilde{v}_2 = v_3 \times v_1$, which has unit length and is orthogonal to $v_1$ and $v_3$. Then, $R = [v_1, \tilde{v}_2, v_3]$.  We will include these details in the revised manuscript.
>
>
> [1] Nashed, Youssef SG, et al. (2021) CryoPoseNet: End-to-end simultaneous learning of single-particle orientation and 3D map reconstruction from cryo-electron microscopy data. Proc IEEE/CVF International Conference on Computer Vision.
>
> [2] Levy, Axel, et al. (2022) CryoAI: Amortized inference of poses for ab initio reconstruction of 3d molecular volumes from real cryo-em images. European Conference on Computer Vision.

---

> > ### Comment · Reviewer_LVtR · 2025-08-04
> >
> > Thank you for the detailed response.
> >
> > Clearly, your method is competitive in terms of runtime but a little expensive in terms of its memory requirements which might hinder its use on lower-end GPUs or large molecules/datasets. I encourage the authors to invest some effort in making the code more memory efficient.
> >
> > In my opinion, none of the other referees raised any serious concerns so I will keep my high scores intact.

---

> > > ### Comment · Reviewer_LVtR · 2025-08-06
> > > **Final Justification copy**
> > >
> > > None of the other reviewers raised any major concerns so I maintain that this is an excellent work. I think the authors did a good job in their rebuttals, including mine.
> > >
> > > Since this is clearly high-quality work in an important and timely field, I strongly support acceptance provided that the authors will release their code.

---

### Official Review · Reviewer_La3P · 2025-07-01

**Clarity:** 4
**Significance:** 4
**Originality:** 4
**Rating:** 6
**Confidence:** 5

**Summary:**

This paper focus on reconstructing heterogeneous biomolecules under complex conditions. Specifically, the authors aim to sovle the 3D reconstruction via cryo-EM images of a single particle under two conditions: (1) The 3D structure of biological particles is often not fixed; due to conformational flexibility, a single biomolecule may adopt multiple structural states across different functional or translational stages, while retaining similar sequences or overall functions. (2) The 2D cryo-EM projections may contain missing parts, caused by partial occupancy, structural incompleteness, or limitations in imaging and particle alignment. To solve these challenges, the authors propose CryoSPIRE, which leverages a part-based Gaussian mixture model to flexibly generte 3D partile structure.

**Questions:**

My main concerns are listed as following:
1. Weakness 1: Part discovery process, whether a different structure between training and test particle will effect the reconstruction performance or not?
2. Weakness 2: the anchor numbers and computational cost.
*********************************************************************************************************

Also I have an extra question, but this won't affect my overall rating of the paper. I believe it is an excellent piece of work.

1. Although I know the authors aim to solve the reconstruction problem in single particle scenario (subtomogram). I'm simply curious about how this method would perform on tomograms with multiple particles.

**Ethical Concerns:**

["NO or VERY MINOR ethics concerns only"]

**Final Justification:**

Thanks for the authors' response. I think the authors have addressed all my concerns. Since I already give the paper the highest rating, I will keep my original score.

**Limitations:**

Yes

**Quality:**

4

**Strengths And Weaknesses:**

Pros:
1. This work aims to address a real-world biological problem, which is quite inspiring.
2. This work is the first to introduce GMM, a generative method commonly used in AI, into the 3D reconstruction process of cryo-EM, which I believe is a highly meaningful contribution. I also believe this approach could be further extended to applications such as 3D cryo-ET generation or structural prediction.
3. The written English of this paper makes it very easy to follow the authors' idea.
4. The anchor-based idea proposed by the authors is very interesting, and I believe this approach could be extended to cryo-ET simulation methods.

Cons:
1. Since the entire part discovery process is based on L2 loss computed from existing projection data, would the performance of missing part reconstruction be affected if the test particle is significantly different from those seen during training—for example, if it has a more complex structure?
2. If a particle has a highly complex structure, would it require a large number of anchors? Would this introduce additional computational overhead?

---

> ### Author Rebuttal · Authors · 2025-07-30
>
> We appreciate that the reviewer finds our work inspiring and our contributions meaningful. In what follows, we answer the questions.
>
> **"...would the performance of missing part reconstruction be affected if the test particle is significantly different from those seen during training—for example, if it has a more complex structure?"... "whether a different structure between training and test particle will effect the reconstruction performance or not?"** Currently, we have focused on the single particle reconstruction case, where the dataset consists of a single particle type. Here, reconstruction is framed as a model fitting to the structure of the same complex which exhibits conformational or compositional variability and the evaluation is based on FSC between output reconstruction and ground-truth structure. As such, for our work to date, we are not testing on out-of-distribution data. Rather we optimize the model from scratch on each new dataset.
>
> **"If a particle has a highly complex structure, would it require a large number of anchors? Would this introduce additional computational overhead?"** For a structure that exhibits more complex dynamics or consists of several compositional parts, we need to accordingly increase the number of anchors/parts. However, as the number of Gaussians is orders of magnitude higher than the number of anchors, the computational overhead to evaluate anchor-specific MLPs to learn rigid part-based motion is not a substantial barrier in our current framework. While the number of anchors is not a significant computational burden, the number of Gaussians does increase the compute cost. Fortunately, one can scale the number of Gaussians significantly, since the image formation model can be parallelized over multiple GPUs straightforwardly.
>
> **"Although I know the authors aim to solve the reconstruction problem in single particle scenario (subtomogram). I'm simply curious about how this method would perform on tomograms with multiple particles."**
> Interesting question. We would like to extend the work to a more general cryoET setting (e.g., with multiple types of particles). However, we expect this would be a major undertaking that would likely require new algorithmic considerations to account for the particularities of cryoET data. As a result, this is unfortunately outside the scope of the current project but would be an excellent direction for future work.

---

> > ### Comment · Reviewer_La3P · 2025-08-01
> >
> > Thanks for the authors' response. I think the authors have addressed all my concerns. Since I already give the paper the highest rating, I will keep my original score.

---

### Official Review · Reviewer_iKnc · 2025-07-02

**Clarity:** 1
**Significance:** 2
**Originality:** 1
**Rating:** 2
**Confidence:** 4

**Summary:**

CryoSPIRE proposes a two-stage cryo-EM workflow:
1. A coarse Gaussian mixture is fitted to 2-D particles; the learned per-Gaussian feature vectors are clustered to identify “parts”.
2. Each part is assigned a rigid transform predicted by an MLP from a latent variable $z$, while finer per-Gaussian offsets and amplitudes model local flexibility.

All parameters and per-image latents are optimized end-to-end with an $\ell_2$ image loss plus Gaussian priors. Experiments on the CryoBench synthetic benchmark and several real-world complexes are reported.

**Questions:**

1. How does performance degrade if the initial orientations are perturbed by 5°–10°?
2. Why is the number of parts set to 32 for IgG and 48 for ribosome? Did you try an information-criterion or cross-validation scheme?
3. Can you report gold-standard FSC, local resolution maps, or atomic‐model fit statistics rather than visual montages?
4. Why are state-of-the-art heterogeneous refinements such as AlphaCryo4D, Bayesian multi-body, and CryoFIRE missing from Table 1?
5. What GPU memory and runtime would be required for a 1 Å map with $G>10^5$ Gaussians and $10^6$ particles?

**Ethical Concerns:**

["NO or VERY MINOR ethics concerns only"]

**Limitations:**

* The method is *not* ab initio; it inherits any alignment and model bias from the initial rigid reconstruction.
* Successful runs require manual tuning of part count, latent size, regularization strength, and MLP width—parameters that vary across datasets with no automatic selection rule.
* Synthetic benchmarks dominate the results; real complexes lack objective resolution or heterogeneity metrics, so claims of biological relevance are unsubstantiated.
* Several competitive heterogeneous-reconstruction methods are omitted, leaving it unclear whether the reported gains hold under fair conditions.
* The paper demonstrates feasibility only on modest particle sets and Gaussian counts; no experiment approaches the data volumes typical of modern cryo-EM studies.

**Quality:**

2

**Strengths And Weaknesses:**

1. The high-level idea—hierarchical Gaussian mixtures with latent-conditioned rigid motions—extends earlier neural-field approaches only marginally. The “part discovery” step is effectively a k-means clustering of per-component features; no principled criterion or unsupervised objective is introduced.
2. The pipeline relies on *pre-estimated* orientations and a good initial rigid map; errors in either propagate unchecked. The latent prior is isotropic Gaussian even for strongly anisotropic motions, and no evidence is provided that the MLP can learn realistic energy landscapes.
3. Quantitative evaluation is almost entirely confined to CryoBench, a synthetic dataset whose ground-truth motions are smooth and part-like. Real-data results are presented only as hand-picked snapshots without resolution statistics or FSC curves. Key baselines (e.g. Bayesian multi-body refinement, AlphaCryo4D) are omitted.
4. Critical hyper-parameters—number of parts, latent dimension, feature-regularization weight—are tuned per dataset with no guidance. Training requires 3–6 hours on a single RTX 2080 for only 10k–20k particles, suggesting prohibitive cost at realistic data scales.
5. The paper is dense, notation changes between sections, and several equations (Eq. 7–9) omit indices needed to implement the method. Figures lack scale bars and resolution annotations, hampering interpretation.

Overall, the weaknesses outweigh the incremental strengths.

---

> ### Author Rebuttal · Authors · 2025-07-30
>
> We thank the reviewer for the feedback. We address the concerns and questions below.
>
> **Hierarchical GMM vs Neural Field (SW1).** Our hierarchical Gaussian mixture model is a completely different representation than previous neural field-based approaches (eg CryoDRGN, DRGNAI) and has key advantages. Unlike neural fields defined on the Fourier domain, it allows explicit parameterization of continuous motion and compositional variability in the real-space (L28-29), facilitating use of spatial priors (L109-111), and it provides substantial gains in rendering efficiency compared to other real-space models like 3DFlex (see supplement Sec C). The use of spatial priors is greatly facilitated through learned Gaussian features that encode information about local structural variability, which would be hard in the Fourier domain. Gaussian splatting exploits similar advantages over neural fields, enabling high-fidelity real-time rendering for 3D/4D scene reconstruction from photos. Such advances are substantial.
>
> **K-means and Part Discovery (SW1).** One key result of our work is that per-Gaussian features learned with a self-supervised reconstruction loss captures useful information about structural variability, despite high levels of noise. This is unexpected and significant, as it facilitates the specification of effective priors for heterogeneity. The specific form of clustering is not key in itself. By way of context, priors for motion-based heterogeneous reconstruction methods like 3DFlex are crucial but also tedious to design in a way that enables fidelity without over-fitting. Our approach is simple by comparison; ie, we learn per-Gaussian features that condition neural networks whose output modulates Gaussian amplitude and location. This deliberate design choice provides the inductive bias driving features to encode local structural heterogeneity, enabling discovery of parts based on a simple algorithm like k-means++, where the number of parts is the only parameter. Further research may reveal better forms of clustering, perhaps incorporating principled biophysics criteria like free energy, but this is outside the scope of this paper.
>
> **Effect of Initialization (SW2, Q1, L1).** CryoSPIRE was not claimed to be ab initio (cf L286 where we note it as future work). Rather, as is common in other methods (eg, cryoDRGN, 3DVA, 3DFlex, RECOVAR), we assume an initial rigid map (which may be very blurry due to motion) and image poses as a starting point. While such initial estimates may be noisy, cryoSPIRE is able to recover from errors in the initial reconstruction, eg, with IgG-1D (Fig 4) the large motion of the Fab domain causes the initial density to be smeared out. Further, fixing initial poses is part of the CryoBench evaluation protocol to ensure that estimated structures are properly aligned with the ground truth references and to ensure fair comparisons with the baselines.
>
> However, per the reviewer’s request, we applied cryoSPIRE and baselines to IgG-1D but with additive noise in pose (uniform over 5 to 10 deg). Results are in the table below, showing that errors in pose cause blur in the 3D reconstructions, degrading the metrics (cf Table 1 in the paper). Nevertheless, CryoSPIRE continues to outperform.
> Method|Per-Conf AUC-FSC
> -|-
> 3DVA|0.243
> 3DFlex|0.231
> CryoDRGN|0.268
> DRGN-AI|0.277
> CryoSPIRE (ours)|0.297
>
> **Isotropic Gaussian Prior, Motion Complexity and Energy Landscapes (SW2).** The sole purpose of the isotropic prior on latent codes is to loosely constrain the scale of the latent embedding. We see no obvious connection between the shape of latent prior and the form of decoded motions per se. In fact, the MLP decodes the latent codes into a variety of motions (see 3D visualizations in the supplement webpage, main.html). We also do not claim that the MLP learns the energy landscape. Rather, its purpose is to learn the mapping between latent codes and the modulation of Gaussian centers and amplitudes to capture conformational and compositional heterogeneity. While some work has aimed to infer the energy landscape (eg AlphaCryo4D or [1]), this is a challenging but distinct problem (see [2]).
>
> **Evaluation Metrics and Datasets (SW3, Q3, L3).** CryoBench (2025) is the only established benchmark for cryo-EM heterogeneity at present. Although synthetic, it contains challenging conformational and compositional variability, based on real cryo-EM studies. Our experimental datasets are widely used to evaluate methods for heterogeneity; as such they enable comparisons with SOTA methods (see supplement Figs 13&14).
>
> CryoBench provides ground-truth and FSC-based performance measures (Per-Conf & Per-Image) to quantitatively compare SOTA methods (Tables 1&3). But there is no established protocol for evaluating heterogeneous reconstruction on experimental data where ground truth is unavailable. Gold-standard FSC is a widely used protocol for homogeneous reconstruction, where one compares two 3D maps from half-sets of the data. For heterogeneity methods, this would produce two different latent conformation spaces without clear correspondences, hence FSC-based comparison is not straightforward.
>
> Gold-standard FSC is even more problematic with methods that use GMM representations.  Specifically, FSC is inherently a measure of consistency of reconstruction and relies on the independence of errors in two reconstructions to provide an estimate of accuracy. This assumption is satisfied with classical cryoEM methods, but GMMs exploit a strong structural prior that induces artificial consistencies at higher frequencies, causing FSC on GMM-based methods to misleadingly over estimate resolution.
> On experimental data, as is common practice (eg, CryoDRGN, RECOVAR), we therefore focus on qualitative comparison. As shown in the supplement pdf and webpage (main.html)  cryoSPIRE outperforms SOTA (3DFlex, 3DVA and CryoDRGN) in recovering fine-grained  detail in highly flexible domains.
>
> **Baseline Methods (SW3, Q4, L4).** We include all baselines in CryoBench (2025), including subspace models, motion-based methods, and neural field methods. The authors of CryoFIRE (2022) led the creation of CryoBench, so we expect cryoFIRE would have been included if it were competitive. Multi-body refinement (2018) is designed for particles with a small number of rigidly moving parts, for which user defined masks must be provided (but are not provided in the CryoBench evaluation protocol). It is not applicable to compositional heterogeneity or more fine-grained continuous conformational deformations. AlphaCryo4D (2022) is focused on a different but related problem of free-energy estimation and uses Relion’s 3D Classification with a large number of discrete conformational states. CryoBench evaluated 3D Classification, the results of which consistently underperform all other methods. That said, we have downloaded the AlphaCryo4D code and will try to run AlphaCryo4D on the CryoBench datasets for completeness.
>
> **Hyperparameter Selection (SW4, Q2, L2).** As reported in Sec 4 (L198), contrary to the review, for all datasets we use default settings of latent dim. $D=4$, regularization weights $\lambda_z=0.1, \lambda_f=0.01$ for feature and latent priors, and $H=32$, $H=128$ hidden dimensions respectively for part-discovery and hierarchical model MLPs.
> Also contrary to the review, the number of parts is not 32 for IgG and 48 for Ribosembly. For IgG-1D and IgG-RL we used 5 or 6 parts (L237). Ribosembly used 8 parts (L241). We allow the user to choose the number of parts and Gaussians, as larger molecules (eg Ribosome) will require more components to reach higher resolutions. At present we visualize the Gaussian features with UMAP (Figs 4-6) from which the number of parts is often clear. While automatic clustering might be useful, cryo-EM practitioners often prefer to choose such parameters, so we don’t view it as critical in practice. Finally, we note our experiment (Table 2, Fig. 9) that explores the effect of different numbers of parts.
>
> **Dataset Sizes, Number of Gaussians and Runtimes (SW4, Q5, L5).** Our datasets are consistent with realistic dataset sizes. IgG-1D and IgG-RL have 100k particles and Ribosembly has >300k particles (L184-187). EMPIAR10076 and EMPIAR10180 have >100k particles each (L191, L195), not 10-20k as stated in the review. The table below reports total runtime, runtime per epoch, number of epochs, and max GPU memory for baselines on CryoBench (3DFlex does not run on Ribosembly). For cryoSPIRE we include the part discovery stage too (hence 15+30 epochs). CryoSPIRE runs significantly faster than competitive methods of CryoDRGN, DRGNAI, and 3DFlex.
>
> We find that an order of $10^4$ Gaussians is sufficient to achieve SOTA quantitative results on CryoBench and high resolution qualitative experimental results. This is also demonstrated in prior work (see [4,5,6] in paper).
> Given the linear image formation model, it is straightforward to share across multiple GPUs to dramatically scale the number of Gaussians. Based on the tables below and for the same number of Gaussians we used (~8k), we estimate running on $10^6$ particles approximately requires 14 min per epoch (630 min in total).
>
> IgG-1D and IgG-RL:
> Method|Total Runtime|Max GPU Mem.|Runtime per Epoch|# Epochs
> -|-|-|-|-
> CryoSPIRE|63 min|5.10 GB|1.40 min|45
> CryoDRGN|304 min|1.34 GB|15.20 min|20
> DRGN-AI|443 min|5.55 GB|4.43 min|100
> 3DVA|37 min|0.75 GB|1.85 min|20
> 3DFlex|725 min|2.12 GB|-|-
>
> Ribosembly:
> Method|Total Runtime|Max GPU Mem.|Runtime per Epoch|# Epochs
> -|-|-|-|-
> CryoSPIRE|405 min|10.60 GB|9.00 min|45
> CryoDRGN|2209 min|1.36 GB|44.18 min|50
> DRGN-AI|1679 min|5.57 GB|16.79 min|100
> 3DVA|120 min|0.75 GB|6.00 min|20
>
> **Notation/Figures (SW5).** We will resolve notational issues and ensure that figures are interpretable.
>
> [1] Tang et al, Ensemble reweighting using cryo-EM particle images, 2023
>
> [2] Evans et al, Counting particles could give wrong probabilities in Cryo-Electron Microscopy, 2025

---

### Decision · Program_Chairs · 2025-09-17

**Decision:**

Accept (poster)

**Comment:**

This paper presents CryoSPIRE, a heterogeneous cryo-EM reconstruction method that models biomolecules with a hierarchical Gaussian mixture representation combined with part discovery. Unlike existing approaches that either rely on generic latent variables or restrict to only conformational or compositional heterogeneity, CryoSPIRE explicitly models both. It achieves state-of-the-art performance on CryoBench and demonstrates promising results on experimental systems. Overall, this is a well-motivated, technically solid, and impactful contribution that advances the state of the art in cryo-EM heterogeneity modeling.